# On Unique Factorization Modules: A Submodule Approach

Sri Wahyuni [1,*,†], Hidetoshi Marubayashi [2,†], Iwan Ernanto [1,†] and I Putu Yudi Prabhadika [3,†]

1 Department of Mathematics, Universitas Gadjah Mada, Sekip Utara, Yogyakarta 55281, Indonesia; iwan.ernanto@ugm.ac.id

2 Department of Mathematics, Naruto University of Education, 748, Nakajima, Takashima, Naruto-cho, Naruto-shi 772-8502, Japan; marubaya@naruto-u.ac.jp

3 Vocational School, Warmadewa University Denpasar, Bali 80235, Indonesia; prabhadika@gmail.com

* Correspondence: swahyuni@ugm.ac.id; Tel.: +62-274-552243

† These authors contributed equally to this work.

**Abstract:** Let $M$ be a torsion-free module over an integral domain $D$. We define a concept of a unique factorization module in terms of $v$-submodules of $M$. If $M$ is a unique factorization module (UFM), then $D$ is a unique factorization domain. However, the converse situation is not necessarily to be held, and we give four different characterizations of unique factorization modules. Further, it is shown that the concept of the UFM is equivalent to Nicolas's UFM, which is defined in terms of irreducible elements of $D$ and $M$.

**Keywords:** unique factorization module; completely integrally closed module; polynomial module



## 1. Introduction

Throughout this paper, $M$ is a torsion-free module over an integral domain $D$ with the quotient field $K$. In [1], the authors introduced a concept of a completely integrally closed module in order to study the arithmetic module theory. $M$ is *completely integrally closed* if for every non-zero submodule $N$ of $M$, $O_K(N) = \{k \in K \mid kN \subseteq N\} = D$.

In Section 2, we define a concept of unique factorization modules (UFMs) as follows. $M$ is a unique factorization module if:

1.    $M$ is completely integrally closed.
2.    Every non-zero $v$-submodule $N$ of $M$ is principal, that is, $N = rM$ for some non-zero $r \in D$.
3.    $M$ satisfies the ascending chain condition on $v$-submodules of $M$.

If $M$ is a UFM, then $D$ is a UFD and $O_K(M) = D$. However, the converse situation is not necessarily to be held (see Example 1). The aim of Section 2 is to provide four different characterizations of UFMs (Theorem 1). Unique factorization modules were first defined by Nicolas in terms of irreducible elements in $M$ and $D$, ([2]) and many interesting results were obtained [2–6]. In Section 3, we show that UFMs in the sense of Nicolas are equivalent to ours, which is proved by using the properties of $v$-submodules (Propositions 2 and 3).

It is well known that $M[x]$ is a UFM over $D[x]$ if $M$ is a UFM [5]. Let $F_v(M[x])$ be the set of all fractional $v$-submodules in $KM[x]$. As an application of Theorem 1, it is shown that $F_v(M[x])$ is naturally isomorphic to $F_v(M) \oplus F_v(M[x])$.

## 2. A Submodule Approach to Unique Factorization Modules

Throughout this paper, $M$ is a torsion-free module over an integral domain $D$ with the quotient field $K$.

**Definition 1.**

1.  *A non-zero D-submodule N of KM is called a fractional D-submodule if there is a non-zero $r \in D$ such that $rN \subseteq M$.*
2.  *A non-zero D-submodule $\mathfrak{a}$ of K is called a fractional M-ideal in K if there is a non-zero $m \in M$ such that $\mathfrak{a}m \subseteq M$.*

Note that we use these concepts [1,7] under the extra conditions $KN = KM$ and $KN^+ = KM$. We denote by $F(M)$ the set of all fractional $D$-submodules in $KM$, and we let $F_M(D)$ be the set of all fractional $M$-ideals in $K$. Let $N \in F(M)$ and $\mathfrak{a} \in F_M(D)$. We define $N^- = \{k \in K \mid kN \subseteq M\}$ and $\mathfrak{a}^+ = \{m' \in KM \mid \mathfrak{a}m' \subseteq M\}$. Then, it easily follows that $N^- \in F_M(D)$ and $\mathfrak{a}^+ \in F(M)$.

For $N \in F(M)$ and $\mathfrak{a} \in F_M(D)$, we define $N_v = (N^-)^+$ and $\mathfrak{a}_{v_1} = (\mathfrak{a}^+)^-$. Then, $N_v \in F(M)$ such that $N_v \supseteq N$, and $\mathfrak{a}_{v_1} \in F_M(D)$ such that $\mathfrak{a}_{v_1} \supseteq \mathfrak{a}$. If $N = N_v$, then we say that $N$ is a fractional *v-submodule* in $KM$. A fractional $M$-ideal $\mathfrak{a}$ is called a $v_1$-ideal ( with respect to $M$) if $\mathfrak{a} = \mathfrak{a}_{v_1}$.

The following properties are easily proved in a similar way as in [1].

Property (A): For any $N \in F(M)$, $N_v = \cap_{N \subseteq kM} kM$, where $k \in K$.

Property (B): The mapping $v : F(M) \longrightarrow F(M)$ given by $v(N) = N_v, N \in F(M)$ is a $\star$-operation on $M$ (see [8], Section 3 for the definition of a $\star$-operation on $M$).

Property (C): Suppose $O_K(M) = \{k \in K \mid kM \subseteq M\} = D$. Then, the mapping $v_1$: $F_M(D) \longrightarrow F_M(D)$ given by $v_1(\mathfrak{a}) = \mathfrak{a}_{v_1}, \mathfrak{a} \in F(D)$ is a $\star$-operation on $D$ (see [8] for the definition of a $\star$-operation on $D$).

Property (D): Let $k \in K$, $\mathfrak{a}$ be fractional $M$-ideal and $N$ be a fractional $D$-submodule. Then:

i.    $(k\mathfrak{a})^+ = k^{-1}\mathfrak{a}^+$.
ii.   $(kN)^- = k^{-1}N^-$.
iii.  $(k\mathfrak{a})_{v_1} = k\mathfrak{a}_{v_1}$.
iv.   $(kN)_v = kN_v$, and $N^- = (N_v)^-$.

In [1], the characterization of completely integrally closed domains is adopted to define a completely integrally closed module.

**Definition 2.** *A torsion-free module M over integral domains D is completely integrally closed if $O_K(N) = \{k \in K \mid kN \subseteq N\} = D$ for every non-zero submodule N of M.*

**Proposition 1.** *([1], Proposition 2.1) M is completely integrally closed if and only if:*

*(1)   Every v-submodule N of M is v-invertible;*
*(2)   $O_K(M) = D$.*

**Proof.** The necessity: Let $N$ be a $v$-submodule of $M$. If $N^-N \subseteq kM$, where $k \in K$, then $M \supseteq k^{-1}N^-N = N^-k^{-1}N$ and $k^{-1}N \subseteq (N^-)^+ = N_v = N$. Thus, $k^{-1} \in O_K(N) = D$, $k^{-1}M \subseteq M$ and so $M \subseteq kM$ follows. It follows that $M \supseteq (N^-N)_v = \bigcap_{N^-N \subseteq kM} kM \supseteq M$ from Property (A). Hence, $M = (N^-N)_v = M$, that is, $N$ is $v$-invertible. It is clear that $O_K(M) = D$.

The sufficiency: Let $N$ be a non-zero $D$-submodule of $M$. First, we prove that $(N^-N)_v = (N^-N_v)_v$. If $N^-N \subseteq kM$, where $k \in K$, then $k^{-1}N^- \subseteq N^-$, and so $k^{-1}N^-N_v \subseteq N^-N_v \subseteq M$, that is, $N^-N_v \subseteq kM$. Hence, $(N^-N_v) \subseteq \bigcap_{N^-N \subseteq kM} kM = (N^-N)_v$ by Property (A).

Let $k \in O_K(N)$, that is, $kN \subseteq N$. Then, $kN_v = (kN)_v \subseteq N_v$ by Property (D). It follows that $M = (N^-N_v)_v = (N^-N)_v \supseteq (N^-kN)_v = k(N^-N)_v = kM$. Therefore, $k \in O_K(M) = D$ by the assumption. Hence, $O_K(N) = D$, that is, $M$ is completely integrally closed. $\square$

**Definition 3.** *M is called a unique factorization module (UFM) if:*

*i.     Every v-submodule N of M is principal, that is, $N = rM$ for some $r \in D$.*
*ii.    $O_K(M) = \{k \in K \mid kM \subseteq M\} = D$.*
*iii.   M satisfies the ascending chain condition on v-submodules of M.*

It can be proved that *M* is a UFM if and only if:

i.     *M is completely integrally closed;*
ii.    *Every v-submodule of M is principal;*
iii.   *M satisfies the ascending chain condition on v-submodules of M,*

which follows Proposition 1.

**Lemma 1.** *Suppose $O_K(M) = D$. Then:*

*(1)    $(\mathfrak{a}M)_v = (\mathfrak{a}_v M)_v$ for every fractional D-ideal $\mathfrak{a}$ in K.*
*(2)    Let $\mathfrak{a}$ be a proper v-ideal of D. Then, $(\mathfrak{a}M)^- = \mathfrak{a}^{-1}$ and $M \supset (\mathfrak{a}M)_v$.*

**Proof.**

(1)    It is clear from Property (B) that $(\mathfrak{a}M)_v \subseteq (\mathfrak{a}_v M)_v$. To prove the converse inclusion, assume $\mathfrak{a}M \subseteq kM$, where $k \in K$, then $k^{-1}\mathfrak{a}M \subseteq M$, and so $k^{-1}\mathfrak{a} \subseteq O_K(M) = D$, that is, $\mathfrak{a} \subseteq kD$. Thus, $\mathfrak{a}_v \subseteq kD$, and $\mathfrak{a}_v M \subseteq kM$ follows. It follows that $\mathfrak{a}_v M \subseteq \bigcap_{\mathfrak{a}M \subseteq kM} kM = (\mathfrak{a}M)_v$ by Property (A), and so $(\mathfrak{a}_v M)_v \subseteq (\mathfrak{a}M)_v$. Hence, $(\mathfrak{a}M)_v = (\mathfrak{a}_v M)_v$.

(2)    We first show that $(\mathfrak{a}M)^- = \mathfrak{a}^{-1}$. It is clear that $\mathfrak{a}^{-1} \subseteq (\mathfrak{a}M)^-$. Conversely, let $k \in (\mathfrak{a}M)^-$, that is, $k\mathfrak{a}M \subseteq M$, so that $k\mathfrak{a} \subseteq D$ by the assumption and $k \in \mathfrak{a}^{-1}$. Hence, $(\mathfrak{a}M)^- = \mathfrak{a}^{-1}$. Suppose $M = (\mathfrak{a}M)_v$. Then, $D = M^- = ((\mathfrak{a}M)_v)^- = (\mathfrak{a}M)^- = \mathfrak{a}^{-1}$ by Property (D), and so $D = \mathfrak{a}^{-1}$, which is a contradiction. Hence, $M \supset (\mathfrak{a}M)_v$.

□

**Definition 4.** *M is called a v-multiplication module if every v-submodule N of M is a multiplication submodule, that is, $N = \mathfrak{n}M$, where $\mathfrak{n} = (N : M) = \{r \in D \mid rM \subseteq N\}$.*

Note that if *D* is a UFD, then every minimal prime ideal is a principal prime (see [8], Theorem 43.14).

**Theorem 1.** *Suppose $O_K(M) = D$. The following conditions are equivalent:*

*(1)    M is a unique factorization module.*
*(2)    M is a v-multiplication module and D is a unique factorization domain.*
*(3)    i.     D is a unique factorization domain, and*
*       ii.    for every prime element p of D, pM is a maximal v-submodule of M, and*
*       iii.   for every v-submodule N of M, $\mathfrak{n} = (N : M) \neq (0)$.*
*(4)    Every v-submodule of M is principal and D is a unique factorization domain.*

**Proof.**

a.     (1) $\Longrightarrow$ (2): It is clear from the definition of UFMs that *M* is a *v*-multiplication module. To prove that *D* is a unique factorization domain, let $\mathfrak{a}$ be a proper *v*-ideal of *D*. Then, $(\mathfrak{a}M)_v$ is a proper *v*-submodule of *M* by Lemma 1, and so $(\mathfrak{a}M)_v = rM$ for some non-unit $r \in D$. It follows that $r^{-1}D = (rM)^- = (\mathfrak{a}M_v)^- = (\mathfrak{a}M)^- = \mathfrak{a}^{-1}$, and so $\mathfrak{a} = \mathfrak{a}_v = rD$.
       Let $\mathfrak{a}_i$ be *v*-ideals of *D* such that $\mathfrak{a}_1 \subseteq \mathfrak{a}_2 \subseteq \dots$. Put $L_i = (\mathfrak{a}_i M)_v = r_i M$ for some $r_i M$, and $\mathfrak{a}_i = r_i D$. Since $L_i \subseteq L_{i+1}$, there is an $n \geq 1$ such that $L_n = L_{n+1}$, that is, $r_n M = r_{n+1}M$. Then, $r_n^{-1} r_{n+1}M = M$, and so since $O_K(M) = D$, $r_n^{-1} r_{n+1} \in D$, that is, $\mathfrak{a}_n = r_n D = r_{n+1}D = \mathfrak{a}_{n+1}$. Hence, *D* is a unique factorization domain.

b.     (2) $\Longrightarrow$ (3): (iii) is trivial since *M* is a *v*-multiplication module. To prove (ii), let *p* be a prime element in *D* and *N* be a *v*-submodule containing *pM*. Then, $\mathfrak{n} = (N :$

$M) \supseteq (pM : M) = Dp$, and $\mathfrak{n}$ is a $v$-ideal of $D$ by Lemma 1. Hence, $\mathfrak{n} = pD$, and so $N = pM = P$ follows. Hence, $pM$ is a maximal $v$-submodule of $M$.

c.　(3) $\Longrightarrow$ (4): Let $N$ be a proper $v$-submodule of $M$. Then, $\mathfrak{n} = (N : M) \neq (0)$, and it is a $v$-ideal of $D$ by (3) (iii) and Lemma 1. Write $\mathfrak{n} = \mathfrak{p}_1^{e_1} \ldots \mathfrak{p}_k^{e_k}$, where $\mathfrak{p}_i$ are different principal prime ideals of $D$ and $e_i \geq 1$ for all $i (1 \leq i \leq k)$. Put $n = e_1 + \cdots + e_k$. If $N = \mathfrak{n}M$, then $N$ is a principal submodule, since $\mathfrak{n}$ is principal. Therefore, we may assume that $N \supset \mathfrak{n}M$ and $N^- = \mathfrak{n}^{-1}\mathfrak{a}$ for some ideal $\mathfrak{a}$ such that $D \supset \mathfrak{a} \supset \mathfrak{n}$. We prove that $N$ is a principal submodule by induction on $n$. If $n = 1$, then $N \supseteq \mathfrak{p}_1 M$ and $N = \mathfrak{p}_1 M$, which is principal by the assumption. Put $P_i = \mathfrak{p}_i M$ for all $i (1 \leq i \leq k)$, which are all maximal $v$-submodules. Suppose that $P_i \not\supseteq N$ for all $i$. Then, $(P_i + N)_v = M$, and so $D = M^- = ((P_i + N)_v)^- = (P_i + N)^- = P_i^- \cap \mathfrak{n}^{-1}\mathfrak{a}$. Thus,

$$D_{\mathfrak{p}_i} = (\mathfrak{p}_i^{-1} \cap \mathfrak{n}^{-1}\mathfrak{a})_{\mathfrak{p}_i} = \mathfrak{p}_i^{-1}D_{\mathfrak{p}_i} \cap \mathfrak{n}^{-1}\mathfrak{a}D_{\mathfrak{p}_i}. \tag{1}$$

If $\mathfrak{n}D_{\mathfrak{p}_i} = \mathfrak{a}D_{\mathfrak{p}_i}$ for all $i$, then $\mathfrak{a} \subseteq \mathfrak{a}D_{\mathfrak{p}_i} \cap D = \mathfrak{p}_i^{e_i} \cap D = \mathfrak{p}_i^{e_i}$ and $\mathfrak{a} \subseteq \mathfrak{p}_1^{e_1} \ldots \mathfrak{p}_k^{e_k} = \mathfrak{n}$, which is a contradiction. There is an $i$, say $i = 1$, such that $\mathfrak{a}D_{\mathfrak{p}_1} \supset \mathfrak{n}D_{\mathfrak{p}_1} = \mathfrak{p}_1^{e_1}D_{\mathfrak{p}_1}$, and so there is an $l$ such that $\mathfrak{a}D_{\mathfrak{p}_1} = \mathfrak{p}_1^l D_{\mathfrak{p}_1}$ with $e_1 > l \geq 0$, since $D_{\mathfrak{p}_1}$ is a discrete rank one valuation domain. Thus, by (1), $D_{\mathfrak{p}_1} = \mathfrak{p}_1^{-1}D_{\mathfrak{p}_1} \cap \mathfrak{p}_1^{l-e_1}D_{\mathfrak{p}_1} = \mathfrak{p}_1^{-1}D_{\mathfrak{p}_1}$, which is a contradiction. Hence, there is a $j$, say $j = 1$, such that $P_1 = \mathfrak{p}_1 M \supset N$, and $\mathfrak{p}_1^{-1}N$ is a $v$-submodule of $M$ with $(\mathfrak{p}_1^{-1}N : M) = \mathfrak{p}_1^{-1}\mathfrak{n} = \mathfrak{p}_1^{e_1-1}\mathfrak{p}_2^{e_2} \ldots \mathfrak{p}_k^{e_k}$. It follows by induction on $n$ that $\mathfrak{p}_1^{-1}N$ is principal, and hence $N$ is a principal submodule as desired.

d.　(4) $\Longrightarrow$ (1): One only needs to prove that $M$ satisfies the ascending chain condition on $v$-submodules of $M$. Let $L_1 \subseteq L_2 \subseteq \cdots \subseteq L_n \subseteq \ldots$ be an ascending chain of $v$-submodules of $M$. Put $L_i = r_i M$ for some non-zero $r_i \in D$ for each $i$. Then, $r_i D = (L_i : M) \subseteq (L_{i+1} : M) = r_{i+1}D$. There is an $n$ such that $r_n D = r_{n+1}D$, since $D$ is a unique factorization domain. Hence, $L_n = L_{n+1}$, and so $M$ satisfies the ascending chain condition on $v$-submodules of $M$.

$\square$

**Remark 1.** *Let $M$ be a UFM and $N$ be a $v$-submodule of $M$. Then, $N$ is a maximal $v$-submodule if and only if $N = \mathfrak{p}M$ for some principal prime $\mathfrak{p}$ of $D$.*

**Proof.** If $N = \mathfrak{p}M$ for some principal prime $p$ of $D$, then it is a maximal $v$-submodule of $M$ by Theorem 1. Conversely if $N$ is a maximal $v$-submodule, then it is a prime submodule (see [7], the proof of Theorem 3.1), and $\mathfrak{n} = (N : M)$ is a prime ideal of $D$. Since $N = \mathfrak{n}M$, it follows from Proposition 1 that $\mathfrak{n}$ is a prime $v$-ideal. Hence, $\mathfrak{n}$ is a principal prime. $\square$

If $M$ is a UFM, then $D$ is a UFD and $O_K(M) = D$. The converse situation is not necessarily to be held.

**Example 1.** *Let $D$ be a UFD, and let $\mathfrak{a}$ be an ideal of $D$ with $\mathfrak{a}_v = D$. Then, $M = \mathfrak{a}$ is not a UFM as a $D$-module.*

**Proof.** It is easy to see that $O_K(M) = D$. Let $p$ be a prime element in $D$ such that $p \in \mathfrak{a}$. Let $L = pD$, a submodule of $M$, and $P = pM$. Then $L \supset P = pM$. It is easy to see that $L^- = p^{-1}\mathfrak{a}$, and so $L_v = (L^-)^+ = (p^{-1}\mathfrak{a})^+ = p\mathfrak{a}^+ = pD = L$. Thus, $P = pM$ is not a maximal $v$-submodule. Hence, $M$ is not a UFM by Theorem 1 part (3). $\square$

See [7], Examples 5.1 and 5.2 for other examples. Example 5.1 is a Krull module and Example 5.2 is a G-Dedekind module, but these are not UFMs.

## 3. The Connection to the Point-Wise Version of the UFM

In [2], Nicolas first defined unique factorization modules in terms of irreducible elements in $D$ and $M$. $M$ is a UFM (a factorial module) in the sense of Nicolas if:

　i.　　Every non-zero element $m$ has an irreducible factorization, that is, $m = r_1 \cdots r_n m'$, where $r_i$ are irreducible elements in $D$ and $m'$ is an irreducible element in $M$.

　ii.　　If $p$ is irreducible in $D$, then $pD$ is a prime ideal.

　iii.　　If $m$ is irreducible in $M$, then it is primitive.

　　　It turns out that $M$ is a UFM in the sense of Nicolas if and only if every irreducible factorization in (i) is unique up to associates (see [2,5]).

　　　The aim of this section is to show that Nicolas's UFM is equivalent to ours by using the properties of $v$-submodules. We refer the reader to [5] and [2] for definitions of irreducible and primitive elements.

**Lemma 2.** *Suppose $O_K(M) = D$. Let $m \in M$ such that $(Dm)^- = D$. Then, $m$ is irreducible.*

**Proof.** Suppose $m = rm'$, where $r \in D$ and $m' \in M$. Then, $D = (Dm)^- = (Drm')^- = r^{-1}(Dm')^-$, and so $(Dm')^- = rD$. Thus, $M = M_v \supseteq (Dm')_v = ((Dm')^-)^+ = (rD)^+ = r^{-1}D^+ = r^{-1}M$ and $r^{-1} \in O_K(M) = D$. Hence, $r \in U(D)$, and so $m$ is irreducible. □

**Lemma 3.** *Suppose $M$ is a UFM in the sense of [2]. Then:*

*(1)　$O_K(M) = \{k \in K \mid kM \subseteq M\} = D$.*

*(2)　If $m$ is primitive, then $(Dm)^- = D$ and $(Dm)_v = M$.*

*(3)　Let $m \in M$ such that $m = rm'$, where $r \in D$ and $m'$ is primitive. Then, $(Dm)^- = r^{-1}D$ and $(Dm)_v = rM$.*

**Proof.** (1)　Let $k \in O_K(M)$ and write $k = ab^{-1}$, where $a, b \in D$ are non-zero. Since $kM \subseteq M$, for a fixed irreducible element $m \in M$, there is an $n \in M$ such that $km = n$, that is, $am = bn$, and we write $n = sm'$ for some $s \in D$ and $m' \in M$, which is irreducible so that $am = bsm'$. Since $D$ is a UFD by ([2], Property 2.2), any irreducible element in $D$ is a prime element. Hence, $a = bsc$ for some unit $c \in D$ by the uniqueness of irreducible factorization, $am = bsm'$. Thus, $k = (bsc)b^{-1} = sc \in D$, and hence $O_K(M) = D$.

(2)　Let $k = ab^{-1} \in (Dm)^-$, where $a, b \in D$ are non-zero. Since $m$ is primitive it follows that $k \in D$ in the same way as in (1), and so $(Dm)^- = D$. Thus, $M = D^+ = ((Dm)^-)^+ = (Dm)_v$.

(3)　$(Dm)^- = (Drm')^- = r^{-1}(Dm')^- = r^{-1}D$ by Property (D) and (2). Hence, $(Dm)_v = ((Dm)^-)^+ = (r^{-1}D)^+ = rD^+ = rM$.
□

**Proposition 2.** *If $M$ is a UFM in the sense of Nicolas, then $M$ is a UFM in our sense.*

**Proof.** $O_K(M) = D$ by Lemma 3. Let $N$ be a proper $v$-submodule of $M$. First, we show that every non-zero element $m \in N$ is not primitive. If $m$ is primitive, then $Dm \subseteq (Dm)_v \subseteq N_v = N$ and so N = M by Lemma 3, which is a contradiction. Thus, every non-zero element $m \in N$ is of the form $m = rm'$, where $r$ is not unit in $D$ and $m'$ is primitive. It follows from Lemma 3 that $N = N_v \supseteq (Dm)_v = rM$, that is, $r \in \mathfrak{n} = (N : M) \neq (0)$. To prove that $N = \mathfrak{n}M$, we assume on the contrary that $N \supset \mathfrak{n}M$. Let $x = sm$ be an element in $N$ but not in $\mathfrak{n}M$, where $s \in D$ and $m$ is primitive. Then again, $N = N_v \supseteq (Dx)_v = sM$ by Lemma 3, and so $s \in \mathfrak{n}$. Thus, $x = sm \in \mathfrak{n}M$, which is a contradiction. Thus, $N = \mathfrak{n}M$. Hence, $M$ is a UFM in our sense by Theorem 1 (2). □

　　　We will prove that the converse is also true, that is, if $M$ is a UFM, then it is a UFM in the sense of [2].

**Lemma 4.** *Let $m$ be an element in a UFM $M$ in our sense. Then:*

*(1)　$m$ is irreducible if and only if $(Dm)_v = M$;*

*(2)　m is irreducible if and only if it is a primitive.*

**Proof.**

(1)　Note that $(Dm)_v = M$ if and only if $(Dm)^- = D$ by Property (D). Therefore, the sufficiency is clear from Lemma 2. The necessity: We assume on the contrary that $M \supset (Dm)_v$. Then, $(Dm)_v = rM$ for some non-unit $r \in D$ and $M = r^{-1}(Dm)_v \ni r^{-1}m$. Thus, there is an element $m_1 \in M$ with $m = rm_1$ and $r \in U(D)$, which is a contradiction. Hence, $(Dm)_v = M$.

(2)　It is well known that any primitive element is irreducible [5]. Suppose $m$ is irreducible and $am' = rm$, where $a, r \in D$ and $m' \in M$. Then, $a^{-1}(Dm')^- = (Dam')^- = r^{-1}(Dm)^- = r^{-1}D$ by (1), and so $a^{-1}D \subseteq r^{-1}D$ since $(Dm')^- \supseteq D$, that is, $aD \supseteq rD$. Therefore, $r = as$ for some $s \in D$ and $am' = asm$. Hence, $m' = sm$ and $m$ is primitive.

　□

**Proposition 3.** *Every UFM in our sense is a UFM in the sense of [2].*

**Proof.** Suppose $M$ is a UFM in our sense. Then, we must prove the following three properties (by the definition):

i.　Every non-zero element $m$ has an irreducible factorization, that is, $m = r_1 r_2 \cdots r_n m'$, where $r_i$ are irreducible in $D$ and $m'$ is irreducible in $M$.

ii.　If $p$ is irreducible in $D$, then $pD$ is a prime ideal.

iii.　If $m$ is irreducible in $M$, then $m$ is primitive.

　　Since $D$ is a UFD by Theorem 1, (ii) is clear and (iii) follows from Lemma 4. To prove statement (i), it is enough to prove that every non-zero element $m$ is of the form $m = rm'$, where $r \in D$ and $m'$ is irreducible in $M$ since $D$ is a UFD. We assume on the contrary that there is a non-zero element $m \in M$ such that $m \neq rm'$ for every $r \in D$ and every irreducible $m' \in M$. Since $m$ is not irreducible, there are $r_1 \in D \setminus U(D)$, and $m_1$ is not irreducible. Therefore, $m = r_1 m_1$, where $r_1 \in D \setminus U(D)$, and $m_1$ is not irreducible. For any natural number $i$, $m_i = r_{i+1} m_{i+1}$, where $r_{i+1} \in D \setminus U(D)$ and $m_{i+1}$ is not irreducible, and $Dm_i \subseteq Dm_{i+1}$. Taking the $v$-operation, we have the ascending chain

$$(Dm)_v \subseteq (Dm_1)_v \subseteq \cdots \subseteq (Dm_i)_v \subseteq \cdots \subseteq M.$$

　　Since $M$ satisfies the ascending chain condition on $v$-submodules of $M$, there is a natural number $n \geq 0$ such that $(Dm_n)_v = (Dm_{n+1})_v$, and so $r_{n+1}(Dm_{n+1})_v = (Dr_{n+1}m_{n+1})_v = (Dm_n)_v = (Dm_{n+1})_v$ by Property (D). Thus, $r_{n+1}^{-1}(Dm_{n+1})v = (Dm_{n+1})_v$, and so $r_{n+1}^{-1} \in O_K((Dm_{n+1})_v) = D$, since $M$ is completely integrally closed. Thus, $r_{n+1} \in U(D)$, which is a contradiction. Hence, every non-zero element $m$ is of the form $m = rm'$, where $r \in D$ and $m'$ is irreducible. Therefore, $M$ is a UFM in the sense of [2].　□

　　We denote by $F_v(M)$ the set of all fractional $v$-submodules in $KM$, where $M$ is a UFM. Let $N$ be a fractional $v$-submodule in $KM$, that is, there is a non-zero $r \in D$ such that $rN \subseteq M$. Then, $M = M_v \supset (rN)_v = rN_v = rN$ by Property (D), and so $rN = sM$ for some $s \in D$ by Theorem 1. Hence, $N = r^{-1}sM$. Conversely, for any non-zero $k \in K$, $kM$ is a fractional submodule in $KM$ and $(kM)_v = kM_v = kM$. Hence, $kM \in F_v(M)$. Hence, $F_v(M) = \{kM \mid 0 \neq k \in K\}$. We define a product "$\circ$" in $F_v(M)$ as follows: $N \circ N_1 = kk_1M$ for $N = kM$ and $N_1 = k_1M$ in $F_v(M)$. Then, $F_v(M)$, endowed with the product $\circ$, is an abelian group generated by the principal primes $\mathfrak{p}M$ and is naturally isomorphic with $F_v(D)$.

**Remark 2.** *Suppose $M$ is a UFM, then:*

*(1)　$F_v(M)$ is an abelian group generated by the principal primes $\mathfrak{p}M$ and is naturally isomorphic with $F_v(D)$.*

*(2)　$F_v(M) = \{kM \mid 0 \neq k \in K\}$.*

The following properties of Krull domain $D$ are more or less known:

(1) $D[x]$ is a Krull domain.

(2) Let $\mathfrak{p}$ be a non-zero ideal of $D[x]$.

    (a) If $\mathfrak{p} \cap D \neq (0)$, then $\mathfrak{p}$ is a minimal prime ideal of $D[x]$ if and only if $\mathfrak{p} = \mathfrak{p}_0[x]$ for some minimal prime ideal $\mathfrak{p}_0$ of $D$. In this case, we say $\mathfrak{p}$ is of type (a).

    (b) If $\mathfrak{p} \cap D = (0)$, then $\mathfrak{p}$ is a minimal prime ideal of $D[x]$ if and only if $\mathfrak{p} = \mathfrak{p}' \cap D[x]$ for some prime ideal $\mathfrak{p}'$ of $K[x]$. In this case, we say $\mathfrak{p}$ is of type (b).

(3) There is a one-to-one correspondence between $\mathrm{Spec}(K[x])$ and $\mathrm{Spec}_0(D[x]) = \{\mathfrak{p} :$ prime $v$-ideals of $D[x] \mid \mathfrak{p} \cap D = (0)\}$, which is given by $\mathfrak{p}' \to \mathfrak{p} = \mathfrak{p}' \cap D[x]$ and $\mathfrak{p} \to K\mathfrak{p}$, where $\mathfrak{p}' \in \mathrm{Spec}(K[x])$ and $\mathfrak{p} \in \mathrm{Spec}_0(D[x])$.

If $M$ is a UFM, then $M[x]$ and $K[x]M[x] = KM[x]$ are both UFMs over $D[x]$ and $K[x]$, respectively ([5], Theorem 6.1 and Result 2.2). Thus, $D[x]$ and $K[x]$ are both UFDs. Thus, $F_v(D[x])$ is an abelian group generated by the minimal prime ideals $\mathfrak{p}_0[x]$ and $\mathfrak{p} \in \mathrm{Spec}_0(D[x])$, where $\mathfrak{p}_0$ are minimal prime ideals of $D$, which are all principal primes in $D[x]$. Hence, $F_v(M[x])$ is an abelian group generated by the $\mathfrak{p}_0[x]M[x]$; and $\mathfrak{p}M[x]$, which are all principal primes of $D[x]$ by Remark 2.

Further, $F_v(KM[x])$ is an abelian group generated by $\mathfrak{p}'M[x]$, where $\mathfrak{p}' \in \mathrm{Spec}\ (K[x])$. It is easy to see that the subgroup of $F_v(M[x])$ generated by the $\mathfrak{p}_0[x]M$ is naturally isomorphic with $F_v(M)$, and the subgroup of $F_v(M[x])$ generated by $\mathfrak{p}M[x]$ is naturally isomorphic with $F_v(KM[x])$. Hence, we have the following remark.

**Remark 3.** *Suppose M is a UFM. Then:*

(1) *$F_v(M[x])$ is an abelian group generated by the $\mathfrak{p}_0[x]M[x]$ and $\mathfrak{p}M[x]$ ($\mathfrak{p}_0[x]$ is of type (a) and $\mathfrak{p}$ is of type (b)).*

(2) *$F_v(M[x])$ is naturally isomorphic with $F_v(M) \oplus F_v(KM[x])$ as abelian groups.*

**Author Contributions:** S.W. (1st author): conceptualization, funding acquisition, investigation, methodology, project administration, and writing—original draft. H.M. (2nd author): conceptualization, investigation, methodology, supervision, and writing—review and editing. I.E. (3rd author): investigation, resources, validation, and writing—original draft. I.P.Y.P. (4th author): methodology, project administration, and resources. All authors have read and agreed to the published version of the manuscript.

**Funding:** This work was partially supported by Universitas Gadjah Mada under Research Grant Year 2021 (Hibah Rekognisi Tugas Akhir).

**Data Availability Statement:** Not applicable.

**Acknowledgments:** This work was initially done while the second author was visiting Universitas Gadjah Mada (UGM) Yogyakarta, Indonesia. He would like to thank algebra's staff at the Department of Mathematics UGM for their kind hospitality during the visit in September–October 2019.

**Conflicts of Interest:** The authors declare no conflict of interest.

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
