# Peer review of "On Unique Factorization Modules: A Submodule Approach"

_axioms, doi:10.3390/axioms11060288_

Round 1

Reviewer 1 Report

see the attachment

Author Response

Dear Reviewer.

Thank you very much for the valuable reviews and comments.

1.The points we revised.

page 1, line 13: a comma between “M” and “OK(N)” is missing.

  • Note: we have added “a comma” between “M” and “OK(N)”

page 3, line 94: remove the period prior to “by”.

  • Note: we have removed “the period” prior to “by”

page 3, line 106: ... of M, and

  • Note: we have added “, and”

page 3, line 118: change “Let” to “let”

  • Note: we have changed “Let” to “let”

page 3, next line of line 121: ... and it is

  • Note: we have changed “and” to “and it is”

page 4, line 129: change “It is only need to” to “One only needs to”

  • Note: we have changed “It is only need to” to “One only needs to”

page 4, line 150: change “another” to “other”

  • Note: we have changed “another” to “other”

2.Revising the references.

Beside revising the above points, following suggestion by the Ms. Lizzy Zhou from Section Managing Editor

“In the reference part, we noticed that you have cited a few of your own
articles. However, as the journal office is now trying to control the number
of self-citations to avoid excessive self-citation rates, please remove the
mentioned citations in your manuscript (no more than 20%)?”

we have deleted the following two references,

[8]: \bibitem{MF} H. Marubayashi and F. Van Oystaeyen, Prime Divisors and Non-commutative Valuation Theory, Lecture Notes in Math. 20059, Springer.1988.

and

[9]: \bibitem{A1} I.E. Wijayanti, H. Marubayashi, I. Ernanto,  Sutopo, Finitely generated torsion-free modules over integrally closed domains, Comm. in Algebra, 8 (48) 2020, 3597-3607, https://doi.org/10.1142/S0219498820501431.

And as the consequences, we make change on

page 2, line 37:

We have deleted reference [9] in “Note that these concepts we use [7], [9] and [10]

under … “ because it is enough with [7] and [10].

page 2, line 49:

We have changed [9] to [2] (R. Gilmer, Multiplicative Ideal Theory, 26,

Marcel Dekker, INC, New York, 1972.)

page 4, line 140:

We have changed (see [9], the proof of Theorem 3.3.) to (see [8], the proof of

Theorem 3.1.)

(Note: [8] previously is [10] of the first version of manuscript as

 consequences of deleting [8] and [9])

3.Changing the style format for the bibliography

Following instruction for authors on https://www.mdpi.com/journal/axioms/instructions we change the style format of the references (the bibliography) as on the revised version on the attachment.

Reviewer 2 Report

Let $D$ be an integral domain and $M$ a torsion-free module over $D$. In 1971 A. M. Nicolas defined when $M$ is a Unique Factorization Module (or UFM). It

implies that $D$ is UFD. The authors of the submitted paper give a different definition of UFM and prove some sets of properties a torsion-free module may have,

each of them equivalent to their definition of UFM. In particular they prove that their definition is  equivalent to the one introduced by Nicolas and studied in the early seventies.

The authors revived the topic in a series of papers. 

As far as I know the paper is correct and the results new, although of limited interest.

Author Response

Thanks a lot for the comments. 

Round 2

Reviewer 1 Report

none